# Technological Advances: *CEBPA* and *FLT3* Internal Tandem Duplication Mutations Can be Reliably Detected by Next Generation Sequencing

**DOI:** 10.3390/genes13040630

**Published:** 2022-04-01

**Authors:** Ratilal Akabari, Dahui Qin, Mohammad Hussaini

**Affiliations:** 1Department of Pathology, Molecular Oncology and Genetics Diagnostics, SUNY Upstate Medical University, Syracuse, NY 13210, USA; rakabari@gmail.com; 2Department of Pathology, Moffitt Cancer Center, Tampa, FL 33612, USA; dahui.qin@moffitt.org

**Keywords:** *CEBPA*, *FLT3*, AML, next generation sequencing (NGS), Internal Tandem Duplication (ITD)

## Abstract

Background: The detection of *CEBPA* and *FLT3* mutations by next generation sequencing (NGS) is challenging due to high GC content and Internal Tandem Duplications (ITDs). Recent advances have been made to surmount these challenges. In this study, we compare three commercial kits and evaluate the performance of these more advanced hybrid-capture and AMP-chemistry based methods. Methods: Amplicon-based TSM 54-Gene Panel (Illumina) was evaluated against hybridization-capture SOPHiA Genetics MSP, OGT SureSeq, and AMP chemistry-based VariantPlex (Archer) for wet-lab workflow and data-analysis pipelines. Standard kit directions and commercial analysis pipelines were followed. Seven *CEBPA* and 10 *FLT3*-positive cases were identified that previously were missed on an amplicon NGS assay. The average reads, coverage uniformity, and the detection of *CEBPA* or *FLT3* mutations were compared. Results: All three panels detected all 10 *CEBPA* mutations and all 10 *FLT3* ITDs with 100% sensitivity. In addition, there was high concordance (100%) between all three panels detecting 47/47 confirmed variants in a set of core myeloid genes. Conclusions: The results show that the NGS assays are now able to reliably detect *CEBPA* mutations and *FLT3* ITDs. These assays may allow foregoing additional orthogonal testing for *CEBPA* and *FLT3*.

## 1. Introduction

Myeloid malignancies predominantly affect the blood and bone marrow and are driven by genomic abnormalities. Detection of mutations in myeloid malignancies is important given their impact on disease diagnosis, prognosis, and/or therapy [1,2]. For example, exon 9 *CALR* mutations in can aid in the diagnosis of myeloproliferative neoplasm such as primary myelofibrosis. Furthermore, the type of *CALR* mutation is relevant since *CALR* type 1–like (c.1099_1150del) and *CALR* type 2–like (c.1154_1155insTTGTC) mutations have differential effects on prognosis [3].

There are more than 100 genes implicated in myelodysplastic syndrome and acute myeloid leukemia [4]. Next-generation sequencing (NGS) allows the testing for large amounts of mutations using massively parallel sequencing with a shorter turn around at lower cost than Sanger sequencing [5]. This makes it feasible to profile a patient’s cancer to determine prognosis or define targeted therapy for optimization of patient outcomes.

However, the NGS technology bears its own complexities and challenges. Some highly important genes clinically are difficult to detect using NGS methods, namely *CEBPA* and *FLT3*. *FLT3* mutations are harbored in 30% of AML and influence prognosis. Mutational impact can be further stratified based on allelic ratio, internal tandem duplication (ITD) size, karyotype, and co-mutations (e.g., *NPM1*). Overall, the presence of *FLT3* mutations is associated with poor survival in AML patients [6,7]. The presence of *FLT3* mutations also determines patient eligibility for therapy with tyrosine kinase inhibitors (eg. Midostaurin, Sorafenib Gilteritinib, etc.) [8]. With regards to *CEBPA*, the presence of biallelic mutations in AML marks a distinct disease entity with a better prognosis compared to patients with single-variant or wild-type *CEBPA* [9].The genetic profile in fact trumps the presence of morphologic dysplasia and these are still classified as AML with biallelic *CEBPA* mutations rather than AML with myelodysplasia-related changes in the setting of dyspoiesis [10].

However, the *CEBPA* gene contains GC rich regions, which are difficult to amplify and sequence [11,12]. *FLT3* mutations typically involve internal tandem duplications (ITD) in the juxtamembrane domain or point mutations/deletions in the tyrosine kinase domain (TKD). ITD mutations are twice as common as TKD mutation and can be large making them challenging to sequence and alignment amplicons of various sizes [12,13]. Because of these challenges and the importance of detecting these mutations clinically, laboratory workflows may often employ duplicate testing of these genes on an orthogonal platform (e.g., Sanger or PCR) to avoid false negative results. This indeed was our practice at our high-volume academic center. However, with advancements in NGS and to increase efficiency in the lab, we explored commercially available solutions to surmount the limitations of our comprehensive genomic profiling (CGP) platform (TruSight Myeloid, Illumina) [14].

NGS kit manufacturers have developed different library preparation and analysis strategies to address challenges in difficult to sequence portions of the genome [15]. In this report, we report the limitations we experienced with the TruSight Myeloid Sequencing panel (Illumina, San Diego, CA USA), and evaluated the performance of enhanced CGP assays (optimized for *FLT3* and *CEBPA*) from three different manufacturers: Myeloid Solution panel (SOPHiA Genetics, Saint Sulpice, Switzerland), SureSeq panel (Oxford Gene Technology, Begbroke, UK), and VariantPlex panel (ArcherDx, Boulder, CO, USA). We assessed wet-lab workflow as well as data analysis performance. The goal of this project was to clinically evaluate these panel to see if they could overcome the known limitations in *CEBPA* and *FLT3* mutation detection that exist with an amplicon-based assay.

## 2. Materials and Methods

### 2.1. Clinical Specimens

Fifteen patient specimens submitted for amplicon based NGS myeloid panel were included in this comparative study. The specimens were selected because they tested positive on a gold-standard single-gene test (fragment analysis or Sanger sequencing) but were negative on an amplicon based NGS assay. Seven of these samples were *CEBPA* positive with a mix of point mutations and insertions/deletions. Ten of these samples were *FLT3* positive cases with ITD up to 107 base pairs.

### 2.2. TruSight Myeloid (TSM) Panel

Library Preparation in TSM Panel employs an amplicon-based approach. The test begins with hybridization of DNA (200 ng measured using NanoDrop) with a multiplexed pool of oligonucleotide probes. The initial correlation between the Qubit and NanoDrop measurements was performed which indicated NanoDrop measured DNA concentrations higher than Qubit (~2.5x). However, NanoDrop quantification (200 ng) yields consistent library complexity (from over 5 years’ worth of the TSM panel data.) Individual oligos in the pool contains a target-specific sequence and an adapter sequence that is used in subsequent PCR amplification. An extension-ligation reaction extends across the target region, followed by ligation to merge the two probes and generate a library of new templates with common ends. The extension-ligation templates are amplified using PCR, which incorporates two unique, library-specific indexes. PCR products are converted to single-stranded fragments and normalized to equimolar concentrations (4 nM). A total of 1.8 pM of pooled libraries were pair-end sequenced on a NextSeq (Illumina, San Diego, CA, USA) with 151 × 2 cycles using the Mid-Output Kit. Sequence data are analyzed using Clinical Genomics Workspace (CGW) from PierianDx (St. Louis, MO, USA). After demultiplexing and FASTQ file generation on the instrument, the CGW pipeline uses NovoAlign Version 3.04.04 software to align the reads against the reference genome (hg19) to create Binary Alignment Map (BAM) files. Novoalign tool trims low-quality calls, removes adaptors, performs Base Quality Score Recalibration (BQSR) and finds optimum alignments for FASTQ reads. Somatic variant callers, FreeBayes and VarScan, then perform variant analysis for the specified regions to generate are Variant Call Format (VCF) files, which contain single nucleotide variants (SNVs) and small insertions/deletions (INDELs). INDEL realignment algorithms are used for the detection of large INDELs in *CALR* and *FLT3* genes.

### 2.3. VariantPlex Myeloid Panel

Libraries were prepared using the VariantPlex protocol (ArcherDx Inc., Boulder, CO, USA) which utilizes Anchored Multiplex PCR (AMP) technology to generate target-enriched sequencing-ready libraries. The input DNA (200 ng measured using Qubit) is first enzymatically fragmented, the ends are blunted, A-tailed, phosphorylated and ligated with half-functional adapters. The adapters contain the universal primer binding sites, index for Illumina instruments and molecular barcodes for deduplication and error correction. The first PCR uses an anchored gene-specific primer 1 (GSP1) which amplifies against P5 primer in the adapter. The second enrichment amplification uses a different nested gene-specific primer 2 (GSP2) to increase amplicon specificity and add read 2 primer binding site. The second primer is hybrid, which contains P7 primer and index 1 region for Illumina instrument. After this cycle, there are two indexes present in every enriched DNA molecule. The data processing was completed via the Archer Analysis platform (ArcherDx Inc.), and the process included FASTQ trimming, read deduplication, genome alignment, and variant detection and annotation. SNPs and small insertions/deletions (indels) of <25 bp are called using FreeBayes and Lofreq. To aid in detection of variants of interest, the ArcherDx variant caller Vision focused on detecting SNPs and small indels of interest by using a targeted VCF file.

### 2.4. Myeloid Solution Panel

The Myeloid Solution Panel by SOPHiA Genetics is based on hybridization-capture chemistry. At least 200 ng of pure DNA (measured using Qubit) is essential for optimal library preparation. DNA is first enzymatically fragmented and then end-repaired and A-tailed using Qiagen QIAseq FX kit. DNA is then ligated with adapter and dual indexes for sample multiplexing later in the process. The cleanup steps are performed to remove non-bound adaptors and size selected (~400 bp) using magnetic beads. A few rounds of PCR amplification are performed to enrich DNA fragments with adaptors. The libraries are cleaned using magnetic beads and quantified, and size verified using TapeStation and Qubit. The libraries are next pooled into a single reaction. Myeloid Solution xGen Lockdown Probes are used to capture the regions of interest. The probe-target duplexes are purified using streptavidin beads protocol. The post-capture amplification is performed to enrich the captured targets. The pooled libraries are again quantified, and size verified. The 1.8 pM of pooled libraries are then subjected pair-end sequencing on a MiSeq (Illumina, San Diego, CA, USA) with 301x2 cycles using Reagent Kit v2 600 cycles cartridge. Sequencing data is analyzed on SOPHiA Data Driven Medicine (DDM) platform. The DDM pipeline uses, PEPPER, proprietary SOPHiA technology, which allows the detection of the *CALR* 52 bp deletions and the *FLT3* ITD up to 177 bp. PEPPER technology is based on a re-alignment algorithm.

### 2.5. SureSeq Panel

Genomic DNA (200 ng measured using Qubit) is enzymatically fragmented using double-stranded New England Biolabs (NEB) Fragmentase to generate fragments of appropriate size (distribution peak at between 150–250 bp). The fragmented dsDNA is repaired with ER enzyme mix to create blunt ends. Simultaneously a 3′ adenine overhang is created for adaptor ligation. High fidelity PCR is used with a few PCR cycles to amplify the library before hybridization and target capture. The amplified library is denatured and captured by biotinylated probes. Then, the hybridized gene targets are bound to streptavidin beads and washed to remove possible off-target DNA. After the capture of targets, PCR is used to add indexes which will identify the sample of each sequence in the NGS run. The dsDNA PCR products then include both index sequences and adaptor sequences. The DNA libraries prepared need to be multiplexed such that each index-barcoded sample is present in the same amounts in the pooled sample. This is predicated on both accurate determination of peak size (bp), performed by TapeStation High-Sensitivity Kit, and accurate determination of library concentration (ng/μL), performed by Qubit High-Sensitivity assay. Data analysis is performed on SureSeq Interpreter software. The Interpreter software uses Qiagen Clinical Insight tool for SNVs and INDELs interpretation.

## 3. Results

The four NGS panels were compared for the genes and/or exons covered, library preparation workflows, depth, unformity and quality of coverage, variant allele fractions and ability to detect variants.

### 3.1. Panel Content Comparision

All panel have some coverage for the following genes relevant to myeloid malignancies: *ASXL1, CALR, CEBPA, DNMT3A, ETV6, FLT3, IDH1, IDH2, JAK2, KIT, KRAS, MPL, NPM1, NRAS, RUNX1, TET2, TP53, U2AF1, WT1*. The panels, however, differ in the target regions for these genes. Some genes are fully covered by all panels, while other genes have coverage only of certain exons (Table 1). To facilitate fair comparison between the panels, a few representative exons from the core myeloid gene list were selected based on kit manufacturer’s claim about the region of interest (ROI) coverage. When multiple exons per gene had very similar coverage (consistent depth among multiple samples and uniform across the ROI) across all four panels), only one of those exons was selected for a comparison, as it would not add value to the comparison.

### 3.2. Workflow Comparision

Each panel in this study uses a different library preparation approach for sequencing the genomic regions of interest. TruSight Myeloid was the only classic amplicon-based panel in this study. ArcherDx VariantPlex uses proprietary AMP chemistry which is similar to amplicon chemistry, but uses a nested PCR-like approach. SureSeq and Myeloid Solution are hybridization capture-based panels primarily distinguished by post-capture amplification in Myeloid Solution library preparation. The ease of use criterion was evaluated based on the number of steps the assay requires and the stage at which the libraries were pooled. SureSeq does not pool the libraries until the denaturation step before loading on the sequencer, which makes it labor intensive because of having to carry each individual library to the end. Overall steps required for TruSight Myeloid panel were the least compared to the other three panels. VariantPlex requires more steps than TruSight Myeloid because of the requirement of the second PCR, but less steps than Myeloid Solution, which requires hybridization, capture and post capture amplification steps. Each library pool was sequenced using either MiSeq or NextSeq sequencer. TruSight Myeloid and VariantPlex libraries were sequenced using 2 × 151 bp cycles on and completed sequencing in 27 h on NextSeq. SureSeq libraries were sequenced using 2 × 151 bp cycles on MiSeq and took 24 h for run completion. Myeloid Solution libraries were sequenced using 2 × 300 bp cycles and completed in 65 h on MiSeq. Workflow comparison is summarized in Table 2.

### 3.3. Depth of Coverage Comparision

Read depth or depth of coverage is the number of reads mapped to a single genomic position after alignment and removal of duplicate reads. The mean read depth is calculated as the total number of aligned bases to the target region divided by the target region size. It indicates how many reads, on average, are aligned at a reference base position. In general, the sensitivity and repeatability of an assay is associated with coverage depth. The read depth of core myeloid genes in each panel is presented on a logarithmic scale in the Figure 1. TruSight Myeloid panel achieved the highest average coverage (18,015), followed by Myeloid Solution (2290), VariantPlex (2217) and SureSeq (692). Comparisons of duplicate reads, on/off target reads, reads without inserts etc. were not within the scope of this project as the purpose of this study was to evaluate manufacturer validated pipelines and analysis filters.

### 3.4. Coverage Uniformity Comparision

Coverage uniformity implies equal distribution of reads along target regions. Uniform coverage reduces the amount of sequencing required to achieve a sufficient coverage depth in targeted regions. NGS assays never achieve full uniformity because some targets are under-sequenced while others are over-sequenced. There are also unavoidable off-target region sequencing. To facilitate fair comparison between the panels, we stipulated that if the exon coverage is 20% lower or 20% higher than the average coverage of the core myeloid genes for that panel, then coverage for that exon was considered to be non-uniform for that exon (Table 3). Among the representative exons selected for comparison, the highest number of uniformly covered exons were in Myeloid Solution (29/39) and SureSeq (28/29). VariantPlex has the least number of uniform exons (5/39) and TruSight Myeloid has 13/39 uniform exons. Coverage uniformity is also evident from Figure 1 where rounder circle with fewer spikes indicates more uniform coverage.

### 3.5. Variant Detection Comparison

Different variant filtering strategies, optimized for each panel by the kit manufacturers, were applied to VCF files as per respective bioinformatics pipelines. Variant allele fraction (VAF) cut-off of 5% was common among the panels. Only clinically relevant variants were chosen for comparison purposes. While the focus of this study was on *CEBPA* and *FLT3*, 47 clinically relevant variants from 15 samples were also included for accuracy comparison (Table 4). Seven *CEBPA* positive and ten *FLT3* positive cases were identified using two criteria: 1. tested positive in single gene test, and 2. tested negative in an amplicon NGS assay. Three *CEBPA* cases had dual *CEBPA* mutations. Overall, in *CEBPA* positive cases, four had point mutations and six had indels. In ten *FLT3* positive cases, the length of ITD was 21 to 107 bp. Sanger Sequencing and fragment analysis orthogonally detected *CEBPA* and *FLT3* gene variants. Myeloid Solution, VariantPlex and SureSeq panels detected all 47 confirmed variants. TruSight Myeloid failed to detect 10 variants, nine of which lay in *CEBPA* or *FLT3*. There was an additional SRSF2 variant, p.P95_R102del, detected by all panels except TruSight Myeloid. While this was not orthogonally confirmed, manual review of the variant in IGV supported it to be a real variant.

### 3.6. BAM Tracks Comparison

In Figure 2, the BAM tracks from all four NGS targeted myeloid panels are loaded in IGV for comparison of the *CEBPA* coverage. TruSight Myeloid panel has the least coverage for *CEBPA* gene because it is an amplicon-based assay and the exon has a GC rich mid-section. VariantPlex panel fully covers the *CEBPA* exon. However, the coverage is not homogenous because the assay takes a nested-PCR like approach. Myeloid Solution has drop in in coverage in the mid-section despite being a hybrid capture assay. However, in this area there is still coverage of about 600x and the peaks are at 5000x. SureSeq panel has the most homogenous coverage among all panels. However, the average coverage is only 700x. While the sequencing depth depends on the sequencing instruments and their capacity, we followed the manufacturer recommended samples per run and per flow cell. Downscaling of the coverage to the factor of the panel with lowest sequencing depth was avoided because it would add bias to the data as we would deviate from the manufacturer recommended sequencing protocols.

In Figure 3, the BAM tracks from all four NGS targeted myeloid panels are loaded in IGV for comparison of the *FLT3* coverage for exon 13–15. Myeloid Solution and SureSeq panels have uniform coverage for all three exons. However, the coverage of SureSeq (790x) is about three orders of magnitude lower than Myeloid Solution (2380x). The coverage of TruSight Myeloid panel appears to more uniform than of VariantPlex for the length of amplicons.

## 4. Discussion

Next generation sequencing technologies refer to a constellation of sequencing methods that share massively parallel sequencing, high throughput, and lower cost [16]. In the clinic, this has allowed for comprehensive genomic profiling to facilitate the timely detection of genetic variants with diagnostic, prognostic, or therapeutic import. In myeloid malignancies, namely AML, the detection of *FLT3* and *CEBPA* alterations are crucial for that very reason [9]. Unfortunately, until recently, limitations in the ability of NGS to detect alterations in these genes required duplicate testing by more sensitive orthogonal method [17]. This prompted academic and private sector efforts in the field to surmount this challenge resulting in various commercially available solutions that claim to reliably detect mutations in this gene [18,19,20]. To increase efficiency in our laboratory workflow by eliminating duplicate testing and to verify vendor claims of accuracy, we performed a head-to-head comparison of TruSight Myeloid (Illumina, San Diego, CA, USA), VariantPlex (Archer), SureSeq (OGT), and Myeloid Solution (SOPHiA) panels in our CAP/CLIA-certified laboratory at a high-volume cancer center.

The hybridization-capture based panels (SureSeq and Myeloid Solution) and proprietary AMP chemistry based VariantPlex panel show promising results for detection of *CEBPA* and *FLT3* variants, all demonstrating 100% sensitivity owing to their unique chemistries and bioinformatics approaches which provided them an advantage over amplicon based TruSight Myeloid panel that detected only 8 of 17 *FLT3* and *CEBPA* variants. All three panels also showed high concordance (100%) detecting 47/47 confirmed variants. This is significant given that detractors of personalized medicine have cited the lack of NGS reproducibility as an argument [21,22]. In this study, we show this not be the case and that reliable NGS results can be procured across different platforms and sequencers based on the current state of technology.

There were differences in coverage metrics between panels, but this did not prevent them from accurately calling the confirmed mutations. Overall, TruSight Myeloid had the deepest coverage but lack of uniformity lends itself to wasted sequencing. The highest uniformity for covered exons were found in the Myeloid Solution panel. All orthogonally confirmed mutations were detected using the three panels being evaluated against TruSight myeloid panel. Analytical sensitivity and specificity and precision studies were not within the scope of this study. We hope to address these details in a forthcoming publication detailing extensive validation work done to assess precision and accuracy of a custom 98-gene panel based on encouraging data from this study. In conclusion, current NGS technologies appear to provide reliable and accurate detection of *CEBPA* and *FLT3* variants surmounting historical challenges with NGS.

## Figures and Tables

**Figure 1 genes-13-00630-f001:**
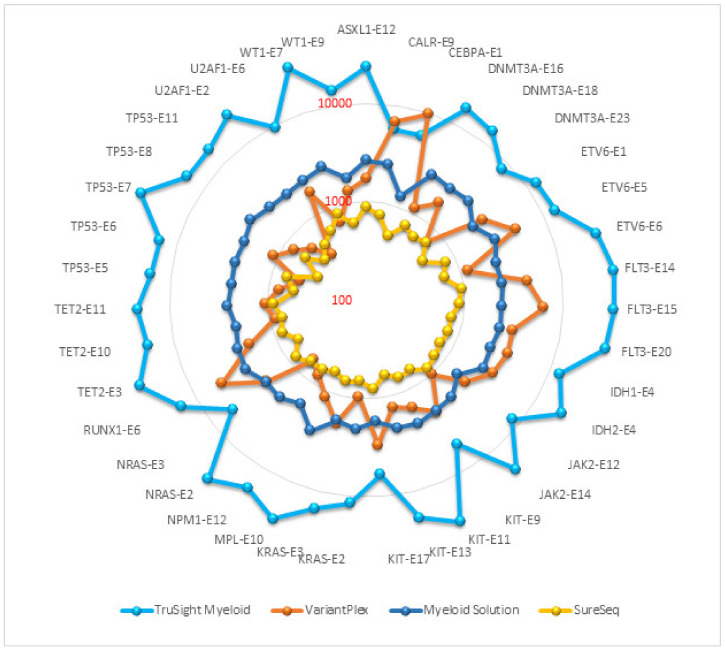
Average coverage of core myeloid genes.

**Figure 2 genes-13-00630-f002:**
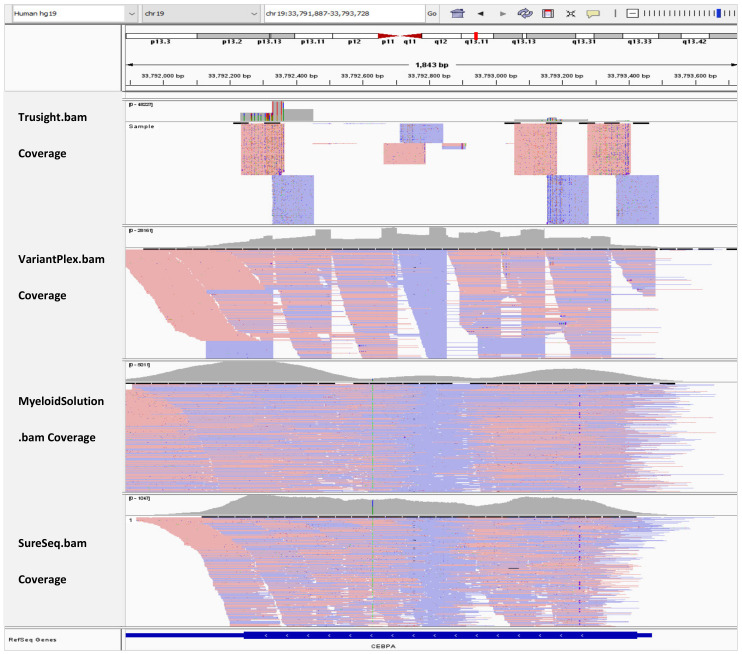
IGV tracks of *CEBPA* exon 1.

**Figure 3 genes-13-00630-f003:**
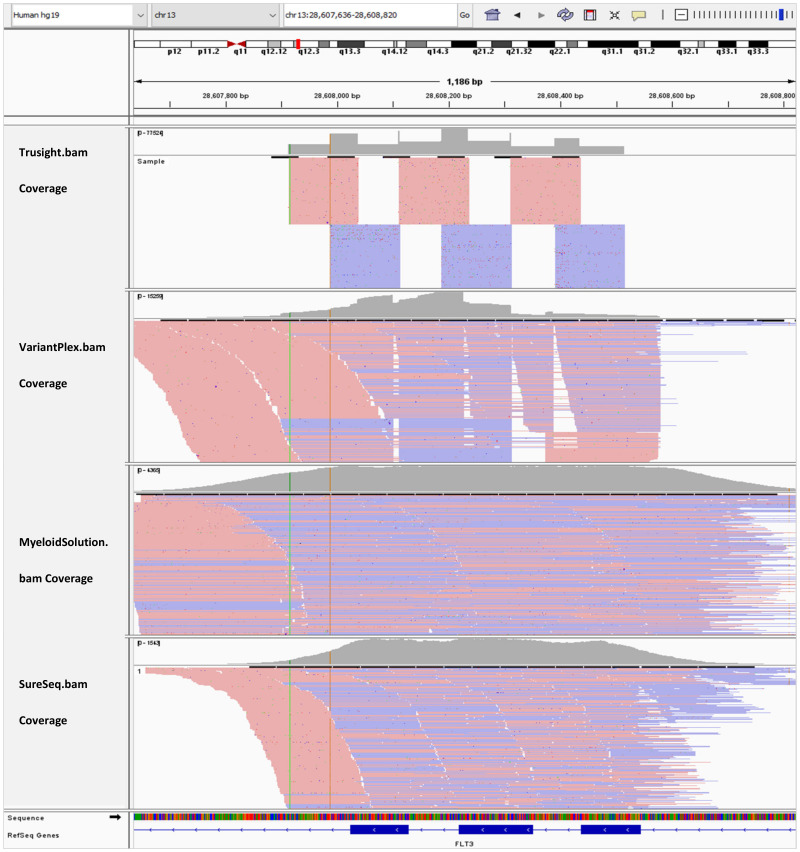
IGV tracks of *FLT3* exon 13–15.

**Table 1 genes-13-00630-t001:** Genes/Exons covered by panels.

Gene	TruSight Myeloid	VariantPlex	SureSeq	Myeloid Solution	Exon/s Selected
*ASXL1*	(12)	(1–13)	(12)	(9,11,12,14)	12
*NPM1*	(12)	(12)	(12)	(11,12)	12
*MPL*	(10)	(10,12)	(10)	(10)	10
*CALR*	(9)	(8,9)	(9)	(9)	9
*IDH1*	(4)	(3,4)	(4)	(4)	4
*IDH2*	(4)	(4,6)	(4–5)	(4)	4
*JAK2*	(12,14)	(12–16,19–25)	(12,14)	all	12,14
*RUNX1*	all	(1–3,5–9)	all	all	6
*KRAS*	(2,3)	(2–4)	(2,3)	(2,3)	2,3
*NRAS*	(2,3)	(2–5)	(2,3)	(2,3)	2,3
*U2AF1*	(2,6)	(2,6,7)	(2,6)	(2,6)	2,6
*TP53*	(2–11)	(1–11)	(2–11)	(2–11)	5–8,11
*TET2*	(3–11)	(3–11)	(2–11)	all	3,10,11
*WT1*	(7,9)	(1–9)	(7,9)	(6–10)	7,9
*FLT3*	(14,15,20)	(8–17,19–21)	(13–15,20)	(13–15,20)	14,15,20
*KIT*	(2,8–11,13,17)	(1,2,5,8–15,17,18)	(2,8–11,13,17)	(2,8–11,13,17,18)	9,11,13,17
*CEBPA*	1	1	1	1	1
*DNMT3A*	all	all	all	all	16,18,23
*ETV6*	all	all	all	all	1,5,6
*ABL1*	(4–6)	(4–10)	(4–6)	(4–9)	NS-ROI *
*ANKRD26*		1 (c.-113-c.-134)			NS-NC **
*ATRX*	(8–10,17–31)	(8–11,17–32)	(8–10,17–31)		NS-NC
*BCOR*	all	(2–15)	all		NS-NC
*BCORL1*	all	all	all		NS-NC
*BRAF*	(15)	(3,10–13,15)	(15)	(15)	NS-ROI
*BTK*		(15)			NS-NC
*CBL*	(8,9)	(2–5,7–9,16)	(8,9)	(8,9)	NS-ROI
*CBLB*	(9,10)	(3,9,10)			NS-NC
*CBLC*	(9,10)	(9,10)			NS-NC
*CCND2*		(5)			NS-NC
*CDKN2A*	all	all	all		NS-NC
*CSF3R*	(14–17)	(10,14–18)	(13–18)	all	NS-ROI
*CUX1*	all	(1–24)			NS-NC
*CXCR4*		(1,2)			NS-NC
*DCK*		(2,3)			NS-NC
*DDX41*		(1–17)			NS-NC
*DHX15*		(3)			NS-NC
*ETNK1*		(3)			NS-NC
*EZH2*	all	(2–20)	all	all	NS-ROI
*FBXW7*	(9–11)	(1–11)	(9–11)		NS-NC
*GATA1*	(2)	(2)	(2)		NS-NC
*GATA2*	(2–6)	(2–6)	(2–6)		NS-NC
*GNAS*	(8,9)	(8–11)	(8–10)		NS-NC
*HRAS*	(2,3)	(2–4)	(2,3)	(2,3)	NS-ROI
*IKZF1*	all	(2–5,7)	all		NS-NC
*JAK3*	(13)	(3,11,13,15,18,19)	(13)		NS-NC
*KDM6A*	all	all			NS-NC
*KMT2A*		(1–36)	(5–8)		NS-NC
*LUC7L2*		(1–10)			NS-NC
*MAP2K1*		(2,3)			NS-NC
*MLL*	(5–8)				NS-NC
*MYC*		(1–3)			NS-NC
*MYD88*	(3–5)	(3–5)	(3–5)		NS-NC
*NF1*		(1–57)			NS-NC
*NOTCH1*	(26–28,34)	(26–28,34,c.*370-c.*380)	(26–28,34)		NS-NC
*PDGFRA*	(12,14,18)	(12,14,15,18)	(12,14,18)		NS-NC
*PHF6*	all	(2–10)	all		NS-NC
*PPM1D*		(6)			NS-NC
*PTEN*	(5,7)	(1–9)	(5,7)		NS-NC
*PTPN11*	(3,13)	(3,4,7,8,11–13)	(3,13)	(3,7–13)	NS-ROI
*RAD21*	all	(2–14)			NS-NC
*RBBP6*		(p.1444, p.1451, p.1569, p.1654, p.1673)			NS-NC
*SETBP1*	(4 partial)	(4 (p.799-p.950))	(4)	(4)	NS-NC
*SF3B1*	(13–16)	(13–21)	(13–16)	(10–16)	NS-ROI
*SH2B3*		(2–8)			NS-NC
*SLC29A1*		(4,13)			NS-NC
*SMC1A*	(2,11,16,17)	(1–25)			NS-NC
*SMC3*	(10,13,19,23,25,28)	(10,13,19,23,25,28)			NS-NC
*SRSF2*	(1)	(1,2)	(1)	(1)	NS-ROI
*STAG2*	all	(2–33)			NS-NC
*STAT3*		(20,21,32)			NS-NC
*U2AF2*		(1–12)			NS-NC
*XPO1*		(15,16,18)			NS-NC
*ZRSR2*	all	all	all	all	NS-ROI

* NS-ROI: Not selected because ROI coordinates were not consistent among the panels. ** NS-NC: Not selected because one or more of the four panels had no coverage.

**Table 2 genes-13-00630-t002:** Workflow comparison.

Library/Sequencing	TruSight Myeloid	VariantPlex	Myeloid Solution	SureSeq
Chemistry	Amplion Sequencing	Anchored Multiplex PCR	Hybridization Capture + post PCR	Hybridization Capture
Preparation Time	1.5 days	2 days	2 days	2.5 days
Customization	Not Available	Available	Available	Available
Instrument	NextSeq	NextSeq	MiSeq	MiSeq
Sequencing Cycles	2 × 151 bp	2 × 151 bp	2 × 300 bp	2 × 151 bp
Sequencing Time	27 h	27 h	65 h	24 h

**Table 3 genes-13-00630-t003:** Coverage uniformity of core myeloid gene (E = exon).

GENE-EXON	TruSight Myeloid	VariantPlex	Myeloid Solution	SureSeq
*ASXL1-E12*	non-uniform	non-uniform	uniform	non-uniform
*CALR-E9*	non-uniform	non-uniform	uniform	uniform
*CEBPA-E1*	non-uniform	non-uniform	non-uniform	non-uniform
*DNMT3A-E16*	uniform	non-uniform	non-uniform	uniform
*DNMT3A-E18*	non-uniform	non-uniform	uniform	uniform
*DNMT3A-E23*	non-uniform	non-uniform	non-uniform	uniform
*ETV6-E1*	non-uniform	non-uniform	uniform	non-uniform
*ETV6-E5*	non-uniform	non-uniform	non-uniform	uniform
*ETV6-E6*	non-uniform	non-uniform	uniform	uniform
*FLT3-E14*	non-uniform	non-uniform	uniform	non-uniform
*FLT3-E15*	non-uniform	non-uniform	uniform	non-uniform
*FLT3-E20*	non-uniform	non-uniform	uniform	uniform
*IDH1-E4*	non-uniform	non-uniform	uniform	uniform
*IDH2-E4*	uniform	non-uniform	uniform	uniform
*JAK2-E12*	non-uniform	uniform	non-uniform	uniform
*JAK2-E14*	uniform	non-uniform	uniform	non-uniform
*KIT-E9*	non-uniform	uniform	uniform	uniform
*KIT-E11*	non-uniform	non-uniform	uniform	uniform
*KIT-E13*	uniform	non-uniform	uniform	uniform
*KIT-E17*	non-uniform	non-uniform	non-uniform	uniform
*KRAS-E2*	non-uniform	non-uniform	uniform	uniform
*KRAS-E3*	uniform	uniform	non-uniform	uniform
*MPL-E10*	non-uniform	non-uniform	non-uniform	uniform
*NPM1-E12*	uniform	non-uniform	non-uniform	uniform
*NRAS-E2*	non-uniform	non-uniform	uniform	uniform
*NRAS-E3*	non-uniform	uniform	uniform	uniform
*RUNX1-E6*	uniform	non-uniform	uniform	uniform
*TET2-E3*	non-uniform	uniform	uniform	uniform
*TET2-E10*	uniform	non-uniform	uniform	uniform
*TET2-E11*	uniform	non-uniform	uniform	non-uniform
*TP53-E5*	uniform	non-uniform	uniform	uniform
*TP53-E6*	uniform	non-uniform	uniform	uniform
*TP53-E7*	non-uniform	non-uniform	uniform	non-uniform
*TP53-E8*	uniform	non-uniform	uniform	uniform
*TP53-E11*	uniform	non-uniform	uniform	non-uniform
*U2AF1-E2*	non-uniform	non-uniform	uniform	non-uniform
*U2AF1-E6*	non-uniform	non-uniform	uniform	uniform
*WT1-E7*	non-uniform	non-uniform	uniform	non-uniform
*WT1-E9*	non-uniform	non-uniform	non-uniform	uniform
Total Uniform	13	5	29	28

**Table 4 genes-13-00630-t004:** Variants comparison.

Sample	Gene	cDNA change	Protein Change	TSM	VP	MS	SS
S1	*FLT3*	c.2503G>T	p.D835Y	0.44	0.46	0.46	0.42
S1	*WT1*	c.1048-2A>C		0.93	0.94	0.91	0.91
S1	*NPM1*	c.859_860insTCTG	p.W288Cfs	0.17	0.29	0.37	0.29
S2	*DNMT3A*	c.2645G>A	p.R693H	0.46	0.45	0.46	0.44
S2	*FLT3*	c.1800_1801ins21	p.D600_L601ins7	0.30	0.34	0.30	0.18
S2	*NPM1*	c.859_860insTCTG	p.W288Cfs	0.10	0.40	0.35	0.44
S2	*U2AF1*	c.101C>A	p.S34Y	0.44	0.43	0.47	0.38
S3	*CEBPA*	c.890G>C	p.R297P	0.39	0.41	0.36	0.44
S3	*FLT3*		ITD 48bp	ND	0.08	0.06	0.09
S4	*CSF3R*	c.2134_2135insTT	p.H739Lfs*91	0.41	0.45	0.47	0.47
S4	*FLT3*	c.1803_1804ins18	p.L601_K602ins6	0.07	0.33	0.43	0.23
S4	*TET2*	c.4524_4525insA	p.Q1510Tfs*68	0.32	0.42	0.47	0.40
S4	*TET2*	c.4716_4717insT	p.P1573Sfs*5	0.25	0.40	0.45	0.47
S4	*U2AF1*	c.101C>T	p.S34F	0.44	0.45	0.42	0.42
S5	*DNMT3A*	c.2285delG	p.G539Afs*17	0.99	0.98	0.96	0.92
S5	*IDH2*	c.515G>A	p.R172K	0.51	0.48	0.48	0.45
S5	*FLT3*		ITD 45bp	ND	0.42	0.51	0.53
S6	*DNMT3A*	c.2645G>A	p.R659H	0.46	0.45	0.46	0.43
S6	*NPM1*	c.860_861insCTGC	p.W288Cfs	0.27	0.39	0.38	0.35
S6	*TET2*	c.3479G>A	p.G1160E	0.50	0.52	0.48	0.44
S6	*ZRSR2*	c.284C>T	p.A95V	0.50	0.48	0.52	0.40
S6	*FLT3*		ITD 75bp	ND	0.44	0.45	0.46
S7	*DNMT3A*	c.2128T>A	p.C710S	0.29	0.14	0.16	0.16
S7	*FLT3*	c.1782_1783ins33	p.F594_R595ins11	0.07	0.11	0.13	0.14
S8	*IDH2*	c.515G>A	p.R172K	0.39	0.39	0.38	0.41
S8	*TP53*	c.838A>G	p.R280G	0.61	0.67	0.58	0.59
S9	*ASXL1*	c.1926_1927insG	p.G646Wfs*12	0.10	0.05	0.09	0.11
S9	*ASXL1*	c.4120G>C	p.V1374L	0.13	0.12	0.11	0.10
S9	*IDH2*	c.419G>A	p.R88Q	0.12	0.10	0.13	0.10
S9	*RUNX1*	c.405G>T	p.R135S	0.11	0.10	0.11	0.09
S9	*FLT3*		ITD 51bp	ND	0.12	0.10	0.15
S9	*SRSF2*	c.284_307del	p.P95_R102del	ND	0.11	0.12	0.10
S10	*CEBPA*	c.901G>A	p.D301N	0.31	0.54	0.55	0.58
S10	*CEBPA*	c.899G>A	p.R300H	0.31	0.55	0.56	0.54
S11	*CEBPA*	c.68_69insC	p.H24Afs*84	0.11	0.06	0.10	0.13
S12	*CEBPA*	c.1020_1021insGC	p.I341Afs	0.36	0.48	0.44	0.41
S12	*CEBPA*	573C>T	p.H191H	ND	0.50	0.48	0.47
S12	*FLT3*		ITD 107bp	ND	0.38	0.40	0.41
S13	*CEBPA*	c.939_940insAAG	p.K313_V314insK	0.25	0.40	0.41	0.37
S13	*CEBPA*	c.247del	p.Q83Sfs*77	0.39	0.42	0.40	0.41
S13	*TET2*	c.895G>T	p.D299Y	0.48	0.51	0.50	0.46
S13	*TET2*	c.3949A>G	p.K1317E	0.48	0.51	0.52	0.45
S14	*DNMT3A*	c.2644C>T	p.R882C	0.34	0.34	0.36	0.39
S14	*TP53*	c.832C>T	p.P258S	0.54	0.49	0.56	0.54
S14	*CEBPA*	c.539delC	p.P180fs*138	ND	0.34	0.33	0.34
S15	*CEBPA*	c.383dupC	p.P128fs*161	ND	0.32	0.32	0.33
S15	*FLT3*		ITD 23bp	ND	0.52	0.55	0.23

ND: Not Detected, TSM: TruSight Myeloid VAF, VP: VariantPlex VAF, MS: Myeloid Solution VAF, SS: SureSeq VAF.

## Data Availability

Not Applicable.

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
