# Peer review of "Technological Advances: CEBPA and FLT3 Internal Tandem Duplication Mutations Can be Reliably Detected by Next Generation Sequencing"

_genes, 2022, doi:10.3390/genes13040630_

Round 1

Reviewer 1 Report

This is a methodological paper and the authors aim to investigate genetic regions that comprise Internal Tandem Duplications (ITDs) and GC reach content in CEBPA and FLT3 genes there were unsuccessfully detected before using next generation sequencing (NGS) approach. The authors compared three kits and tested performance of hybrid-capture and AMP-chemistry based methods. Amplicon-based TSM 54-Gene Panel form Illumina was  compared to hybridization-capture SOPHiA Genetics MSP, OGT SureSeq, and AMP chemistry-based VariantPlex (Archer) for lab workflow and data management. Every method, was described in detail allowing for reproducibility.  Also differences between panels were clearly shown including targets description (coverage) in gene panels. Crucial sequencing parameters, important for perspective of practical usage were also elucidated and encompassed workflow, depth of coverage uniformity and variant detection (accuracy). Finally, generated BAM files were shown using in IGV, and differences between panels were presented in a graphical plot. The only regret is that authors decided to evaluate the analytical sensitivity and specificity of panels in a separate paper.  I think it would be better to have it one paper, also maybe a little bit more compacted. No other issues. The Paper is worth to published and I’m sure it will be helpful for units that consider and maybe hesitate which approach to choose. Also helpful for researchers who are already using on of presented  method, to reconsider another option.

Author Response

Thank you for your kind words and positive adjudication regarding the manuscript. We agree that additional data concerning precision studies (analytic sensitivity and specificity) would have been of  additional value, but these were beyond the scope of this study. Moreover, since the analytic sensitivity and specificity of NGS has been abundantly published upon, the addition of this type of data on a small number (n=15) of cases we feel would be of limited value.

Reviewer 2 Report

The article is well written, no plagiarism was detected. My main concern is the following: 

Major:

What is the added value of the current paper ?? At least three papers previously investigated this topic (See below). The authors need to show the added value of their paper..

A comparative study of next-generation sequencing and fragment analysis for the detection and allelic ratio determination of FLT3 internal tandem duplication. Jin Ju Kim, Diagnostic Pathology (2022) 

Next-generation sequencing of FLT3 internal tandem duplications for minimal residual disease monitoring in acute myeloid leukemia. Oncotarget, 2015 Sep 8;6(26):22812-21.  doi: 10.18632/oncotarget.4333.

Hybridization capture-based next-generation sequencing reliably detects FLT3 mutations and classifies FLT3-internal tandem duplication allelic ratio in acute myeloid leukemia: a comparative study to standard fragment analysis. Mod Pathol. 2020 Mar;33(3):334-343. doi: 10.1038/s41379-019-0359-9

Minor:
 Genes symbols need to be italicized all over the text, including the abstract. 

Author Response

Comments and Suggestions for Authors

The article is well written, no plagiarism was detected. My main concern is the following:

Major:

What is the added value of the current paper ?? At least three papers previously investigated this topic (See below). The authors need to show the added value of their paper..

A. A comparative study of next-generation sequencing and fragment analysis for the detection and allelic ratio determination of FLT3 internal tandem duplication. Jin Ju Kim, Diagnostic Pathology (2022)

B. Next-generation sequencing of FLT3 internal tandem duplications for minimal residual disease monitoring in acute myeloid leukemia. Oncotarget, 2015 Sep 8;6(26):22812-21. doi: 10.18632/oncotarget.4333.

C. Hybridization capture-based next-generation sequencing reliably detects FLT3 mutations and classifies FLT3-internal tandem duplication allelic ratio in acute myeloid leukemia: a comparative study to standard fragment analysis. Mod Pathol. 2020 Mar;33(3):334-343. doi: 10.1038/s41379-019-0359-9

Author Response:

We appreciate the reviewer’s concern and desire for clarification for the additional value our study offers to the scientific cannon. However, we provide substantiation of additional value below:

  1. First and most importantly, we report on the ability of NGS panels to now reliably detect CEBPA which is not addressed in any of the above studies. This is clearly an added value from the other studies. Since CEBPA, NPM1 and FLT3 testing are all of paramount importance in cytogenetically-normal AML genomic profiling at diagnosis, we deem our study as important given that it substantiates the ability to completely do the profiling by NGS alone without reflexing to one (or more) additional tests.
  2. I myself trained at Washington University and one of my colleagues, David Spencer, showed early on in this field that FLT3 ITD can in fact be detected by NGS as far back as 2013 (PMID: 23159595). However, clinical implementation of these assays and their performance in the real world turned out to be a departure from what was shown in the research world. The limitations of FLT3 detection in clinically assays was a known pitfall for laboratory directors and resulted in orthogonal testing (such as in our lab using one of the most widely available kits for myeloid sequencing, TSM 54, on the market). Herein, lies the novelty of our study. The added value of this study is that it shows how the use of commercially available kits using standard manufacturer defined pipelines (not custom-designed research protocols) can now be deployed to reliably detect CEBPA and FLT3. This is an additional value.
  3. The provided studies do not compare more than one widely-available commercial kits for profiling. Since most laboratories do not have the ability to design and manufacture their own panels, most centers rely on commercially-available kits for this purpose. In this study, we compare and provide data regarding some of the leading panels in the market. This is not found in the listed studies.
    1. Studies B and C use Agilent probes for NGS (HaloPlex Capture and SureSelectX). This does not provide information for other widely-used kits that we have provided in our study. This will be very helpful to the readership in terms of practicality.
    2. Study A apparently used an in-house 497 gene panel in Korea. This would not be something that could be easily replicated or deployable here in the US.
    3. In Study B, results for ITD detection were concordant for 17 patients (85%) between fragment analysis, NGS analysed with Pindel and NGS analysed with doMreps. In particular, three samples Pindel and doMreps failed to detect the same ITD as fragment analysis. For the most frequently mutated gene in AML, and one that is druggable to wit, a 85% detection rate would be inadequate clinically. Our study showed 100% detection.
  4. Even though a handful of studies may have been published (mostly recently) showing that FLT3 can be detected by NGS, there is need for further publications independently from various academic centers and labs to validate this proposition and create trust in the scientific community regarding reliable detection. We feel that our study is a contribution to the larger body of literature in this regard and needed to create a ground swell.

Minor:

 Genes symbols need to be italicized all over the text, including the abstract.

Author’s Response:

This has been done now. Thank you.

Reviewer 3 Report

In the manuscript entitled “Technological Advances: CEBPA and FLT3 Internal Tandem Duplication Mutations Can be Reliably Detected by Next Generation Sequencing” by Ratilal Akabari and colleagues, authors compare four commercial NGS kits and evaluate their performance in detecting alterations in the CEBPA and FLT3 genes in the samples of 15 patients diagnosed with AML. Although the research topic is timely and highly relevant, study is limited by the specter of analysis that authors performed as well as the conclusions that cannot always be sustained by the presented results.

Major comments:

  1. What was the reason to use different DNA quantification technique for TSM panel compared to capture-based panels? NanoDrop does not quantify only double stranded DNA, like Qubit, which is why this method may cause overestimation of the DNA quantity, and subsequently influence library complexity. Authors should examine the correlation between the measured concentration with Qubit and NanoDrop to ascertain that there were no big discrepancies, which could influence library complexity.
  2. In the section 3.1, lines 167 and 168, authors should explain what kind of criteria they applied in order to avoid bias when selecting exons for comparison.
  3. In the table 1 authors should add a column with the information about the exons selected for comparison for each gene.
  4. In the section 3.2, authors make comparison between different protocols and rank each of the method based on “ease of use”. However, authors do not describe how this ease of use was evaluated and to me it appears that this decision, as well as ranking of different protocols, was subjective. Authors should either make a quantitative scale to objectively judge “ease of use” of each library prep protocol based on the defined set of parameters, or alternatively, conduct a survey among technicians performing these experiments in order to quantify their subjective opinion on the “ease of use” for a defined set of parameters.
  5. In addition to average depth, other sequencing metrics could also be informative. For example, authors could compare number of duplicate reads, number of reads mapping on target and off target, number of unmapped reads, number of reads that do not contain insert and etc. Adding these layers of information could really aid in making conclusions about performance of different methods.
  6. In addition to showing which method was successful in detecting mutations, authors should add information about mutation allele frequency (MAF) and how the estimated MAF correlated between different methods. Furthermore, authors should add more information about detection threshold of each method, which is particularly important when it comes to evaluating mutations with very low MAF.
  7. In the section 3.6, authors made conclusions, which by my opinion cannot be supported by the presented results. This was in particular the case for the results obtained from the SureSeq panel, where authors present the coverage of only 700x and 790x for CEBPA and FLT3 genes, respectively, and compare it with the coverage obtained from Myeloid Solutions. In fact, average coverage from the SureSeq panel was 692 which means that the results shown clearly indicate that the coverage in these regions are similar to the one encompassing whole targeted region. Comparison with Myeloid Solutions in this case is inadequate, since the Myeloid Solutions had on average 3x higher coverage. If the authors would downscale coverage of the Myeloid Solutions 3x (going from average 2290 to 763) than the coverage in FLT3 gene would be 793, which is similar to what was observed with SureSeq panel. Because the sequencing depth largely depends on the sequencing platform and their capacity, for these kind of comparisons authors should randomly downscale samples with high sequencing depth to the factor of the samples with lower sequencing depth and then make comparison regarding coverage uniformity. Authors should also consider making density diagrams instead of IGV tracks for figures 2 and 3 and overlay the curves for each method. This would make interpretation much easier. IGV tracks can still be presented as supplements.
  8. Discussion is very short and could benefit from giving more information about advantages and shortfalls of different capture-based techniques. Current conclusion is that the capture-based techniques are superior to one amplicon-based technique, but with the amount of data generated in this study authors could say more about pros and cons of different methods.

Minor comments:

  1. Authors should make short, 1-2 sentences introduction in the beginning of the result section where they would in short recapitulate what they did. Many readers are skipping introduction and method section, so mentioning what was the aim and what was done would improve the overall quality of the manuscript.
  2. Gene names should be written in italic
  3. Introduction, line 31 – it appears that authors omitted name of the gene here

Author Response

Comments and Suggestions for Authors

In the manuscript entitled “Technological Advances: CEBPA and FLT3 Internal Tandem Duplication Mutations Can be Reliably Detected by Next Generation Sequencing” by Ratilal Akabari and colleagues, authors compare four commercial NGS kits and evaluate their performance in detecting alterations in the CEBPA and FLT3 genes in the samples of 15 patients diagnosed with AML. Although the research topic is timely and highly relevant, study is limited by the specter of analysis that authors performed as well as the conclusions that cannot always be sustained by the presented results.

Major comments:

Observational. All Qubit.

What was the reason to use different DNA quantification technique for TSM panel compared to capture-based panels? NanoDrop does not quantify only double stranded DNA, like Qubit, which is why this method may cause overestimation of the DNA quantity, and subsequently influence library complexity. Authors should examine the correlation between the measured concentration with Qubit and NanoDrop to ascertain that there were no big discrepancies, which could influence library complexity.

Part of the reason for this was historical. Our CAP/CLIA TSM 54 panel was validated using a certain standard protocol conditions and these persisted and were not altered for the sake of this study. In addition, we have added the following clarification (P2; lines 86-89):

“The initial correlation between the Qubit and NanoDrop measurements was performed which indicated NanoDrop measured DNA concentrations higher than Qubit (~2.5x). However, NanoDrop quantification (200 ng) yields consistent library complexity (from over 5 years’ worth of the TSM panel data.)”

In the section 3.1, lines 167 and 168, authors should explain what kind of criteria they applied in order to avoid bias when selecting exons for comparison

The criteria we used has been added (P4; lines 172-176):

“To facilitate fair comparison between the panels, a few representative exons from the core myeloid gene list were selected based on kit manufacturer’s claim about the region of interest (ROI) coverage. When multiple exons per gene had very similar coverage (consistent depth among multiple samples and uniform across the ROI) across all four panels), only one of those exons was selected for a comparison, as it would not add value to the comparison.”

Furthermore, the Inclusion of too many exons would complicate visual representation on the spider plot without adding addition value for the reasons stated above.

Using stated ROIs ensure that breadth of coverage was the same, and we were performing a fair comparison.

In the table 1 authors should add a column with the information about the exons selected for comparison for each gene.

This has been added to Table 1.

In the section 3.2, authors make comparison between different protocols and rank each of the method based on “ease of use”. However, authors do not describe how this ease of use was evaluated and to me it appears that this decision, as well as ranking of different protocols, was subjective. Authors should either make a quantitative scale to objectively judge “ease of use” of each library prep protocol based on the defined set of parameters, or alternatively, conduct a survey among technicians performing these experiments in order to quantify their subjective opinion on the “ease of use” for a defined set of parameters.

We agree that to a certain degree this ranking is subjective in nature. Inter-technician could not be performed given that there was only one technician doing the validation runs. However, our approach was not entirely arbitrary and there was a methodical reasoning behind our assessment. This has been added to the text (P6; 186-193):

“The ease of use criterion was evaluated based on the number of steps the assay requires and the stage at which the libraries were pooled. SureSeq does not pool the libraries until the denaturation step before loading on the sequencer, which makes it labor intensive because of having to carry each individual library to the end. Overall steps required for TruSight Myeloid panel were the least compared to the other three panels. VariantPlex requires more steps than TruSight Myeloid because of the requirement of the second PCR, but less steps than Myeloid Solution, which requires hybridization, capture and post capture amplification steps.”

In addition to average depth, other sequencing metrics could also be informative. For example, authors could compare number of duplicate reads, number of reads mapping on target and off target, number of unmapped reads, number of reads that do not contain insert and etc. Adding these layers of information could really aid in making conclusions about performance of different methods.

We agree that provision of these additional metrics may be useful to the readership. However, they would require significant bioinformatic work, delay submission and did not fall in the in the immediate scope of paper whose primary intent was to show that FLT3 and CEBPA can reliably detected now using commercially available kits. This has been clarified (P7; lines 209-212)

Comparisons of duplicate reads, on/off target reads, reads without inserts etc. were not within the scope of this project as the purpose of this study was to evaluate manufacturer validated pipelines and analysis filters

In addition to showing which method was successful in detecting mutations, authors should add information about mutation allele frequency (MAF) and how the estimated MAF correlated between different methods. Furthermore, authors should add more information about detection threshold of each method, which is particularly important when it comes to evaluating mutations with very low MAF.

This is a good point. Detection threshold has been added (P9; lines 232-233):

Variant allele fraction (VAF) cut-off of 5% was common among the panels.

Furthermore, detected vs. not detected has been replaced with the MAF values in Table 4 (P9)

In the section 3.6, authors made conclusions, which by my opinion cannot be supported by the presented results. This was in particular the case for the results obtained from the SureSeq panel, where authors present the coverage of only 700x and 790x for CEBPA and FLT3 genes, respectively, and compare it with the coverage obtained from Myeloid Solutions. In fact, average coverage from the SureSeq panel was 692 which means that the results shown clearly indicate that the coverage in these regions are similar to the one encompassing whole targeted region. Comparison with Myeloid Solutions in this case is inadequate, since the Myeloid Solutions had on average 3x higher coverage. If the authors would downscale coverage of the Myeloid Solutions 3x (going from average 2290 to 763) than the coverage in FLT3 gene would be 793, which is similar to what was observed with SureSeq panel. Because the sequencing depth largely depends on the sequencing platform and their capacity, for these kind of comparisons authors should randomly downscale samples with high sequencing depth to the factor of the samples with lower sequencing depth and then make comparison regarding coverage uniformity. Authors should also consider making density diagrams instead of IGV tracks for figures 2 and 3 and overlay the curves for each method. This would make interpretation much easier. IGV tracks can still be presented as supplements.

We appreciate the suggestion to perform downscaling and compare results. However, for the purposes of our study we do not deem that this would be the best approach because:

  1. Each panel has been optimized by the manufacturer to yield reliable results. The balance area also balanced based on number of genes. Our intent was not to be completely theoretical. Inherent difference in workflows (such as post-capture amplification) are expected to result in differences in reads generated, duplicates, etc. What we really wanted to see was how each panel performs when using manufacturer provided SOPs. This reflects practically how these panels would be deployed in the real world the their ensuing results.
  2. Downscaling bioinformatically, while correcting for equalization in one sense, is a stochastic process and could introduce bias in the results.

We opted not to provide density diagrams as they would be redundant with the spider plot. IGV figures were provided as they allow for visualization of raw data which is something that the molecular pathologists in the readership would appreciate. 

We added the following to the text for further clarity (P10; lines 260-262):

“Downscaling of the coverage to the factor of the panel with lowest sequencing depth was avoided because it would add bias to the data as we would deviate from the manufacturer recommended sequencing protocols.”

Discussion is very short and could benefit from giving more information about advantages and shortfalls of different capture-based techniques. Current conclusion is that the capture-based techniques are superior to one amplicon-based technique, but with the amount of data generated in this study authors could say more about pros and cons of different methods.

This has now been added to section 3.2 (P6; lines 186-193). We did not repeat this is the discussion section for the sake of brevity and to avoid redundancy.

Minor comments:

Authors should make short, 1-2 sentences introduction in the beginning of the result section where they would in short recapitulate what they did. Many readers are skipping introduction and method section, so mentioning what was the aim and what was done would improve the overall quality of the manuscript.

We have added the following (P4; lines 163-165):

“The four NGS panels were compared for the genes and/or exons covered, library preparation workflows, depth, unformity and quality of coverage, variant allele fractions and ability to detect variants.”

Gene names should be written in italic

This has been done.

Introduction, line 31 – it appears that authors omitted name of the gene here

You are right. We have added (P1; line 31)

 “..CALR.”

Round 2

Reviewer 2 Report

The reviewer proposes that the authors discuss these points in the discussion section. This will increase the reader's interest in their paper

Reviewer 3 Report

Authors answered most of my major comments and addapted the manuscript accordingly. 

Although I understand the point authors were trying to make in the manuscript, i.e., compare different NGS methods using comercially optimized and validated pipelines, I do believe that deeper examination of the current data would greatly add to the overal quality of the manuscript. In that respect, examining the quality, sequencing metrics and analytic sensitivity of each approach would be of great interest to the readers and would allow for the current conclusion of the manuscript to be expanded beyond capture-based methods are better than amplicon-based. In that regard, I feel like authors missed an oportunity to create a manuscript that could be used by diagnostic laboratories to make informed decission about the best approach to use for their needs. Neverthless, current version of the manuscript still offers enoungh interesting observations to warant publication.